# Micronutrient Deficiencies in Medical and Surgical Inpatients

**DOI:** 10.3390/jcm8070931

**Published:** 2019-06-28

**Authors:** Mette M Berger, Olivier Pantet, Antoine Schneider, Nawfel Ben-Hamouda

**Affiliations:** Service of Adult Intensive Care Medicine and Burns, Lausanne University Hospital (CHUV), BH 08.612, Rue du Bugnon 46, 1011 Lausanne, Switzerland

**Keywords:** iron, copper, selenium, zinc, thiamine, vitamin B12, obesity, inflammation, enteral nutrition

## Abstract

Inpatients are threatened by global malnutrition, but also by specific micronutrient (i.e., trace element and vitamins) deficiencies that frequently are overseen in the differential diagnosis of major organ dysfunctions. Some of them are related to specific geographic risks (iodine, iron, selenium, zinc, vitamin A), while others are pathology related, and finally many are associated with specific feeding patterns, including low dose enteral feeding. Among the pathologies in which laboratory blood investigations should include a micronutrient outwork, anemia is in the front line, followed by obesity with bariatric surgery, chronic liver disease, kidney disease, inflammatory bowel disease, cardiomyopathies and heart failure. The micronutrients at the highest risk are iron, zinc, thiamine, vitamin B12 and vitamin C. Admission to hospital has been linked with an additional risk of malnutrition—feeding below 1500 kcal/day was frequent and has been associated with a structural additional risk of insufficient micronutrient intake to cover basal needs. Although not evidence based, systematic administration of liberal thiamine doses upon admission, and daily complementation of inpatients’ food and enteral feeding solutions with multi-micronutrient tablets might be considered.

## 1. Introduction

Malnutrition includes a wide spectrum of conditions [1] that may affect energy, substrates and micronutrients (i.e., trace elements and vitamins) to variable degrees. Deficiency is defined as a lack, or shortage of a specific micronutrient that is essential for the proper growth and metabolism of a human: 11 trace elements and 13 vitamins qualify as essential in humans [2,3].

Disease related malnutrition has shown to be frequent upon admission [4]. The proportion of patients that have been admitted to hospital in a poor nutritional status varies between nearly zero in trauma and up to 50% in oncologic pathologies [5,6], and has an important impact on costs [6]. Malnutrition affects all ages from pediatric [7] to geriatric admissions [8]. Elderly subjects constituted an important, increasing part of our population with specific physiopathologic characteristics [9]. In developing countries, socio-economic factors will generate specific deficiencies [10], which will also be present in migrant populations [11]. Further, some pathologies threaten more, particularly the micronutrient status. Finally, the geographic place of living in the world, including some western countries and the local soil characteristics, will have their own impact [12,13]. Iodine and iron deficiencies are the most well-known. The below text will review some of these undermining deficiencies.

The definition of deficiency requires knowing the specific needs of diseased patients—this information is generally not available. Micronutrient requirements have been determined for the healthy population with age specificities [3]. They are called dietary reference intakes (DRI), a concept that has replaced the recommended daily allowance (RDA). There is minimal data that exists about the needs during disease. Therefore, the present review, which is based on searches in PubMed and the Cochrane library, is focused on conditions in adults that have been best documented, such as elderly patients, and some selected conditions. The issue of refeeding syndrome in hospitalized patients is be given particular attention.

## 2. Undermining Micronutrients Deficiencies

The causes of deficiency can globally be divided into three: Low availability, reduction of intakes, and malabsorption. Some geographic soil and nutrition specificities have threatened the entire populations’ micronutrient status. This knowledge is not new [2], and the below text provides a brief summary of their impact on acute diseases.

Micronutrient deficiencies have been called the hidden hunger [14], as they are determining and aggravating factors for one’s health status and quality of life. The close relation between fetal malnutrition and the development of chronic non-communicable diseases later in life has been repeatedly confirmed [10,15]. Vitamin A (in developing countries), iodine, and iron deficiencies (worldwide) are the most important in terms of global public health [16]. The below deficiencies have affected patients before their hospital admission and should be integrated in the differential diagnosis of the acute condition.

### 2.1. Iodine

Deficiencies early in life impairs cognition and growth. Iodine status is also a key determinant of thyroid disorders in adults. Iodine deficiency disorders have affected 740 million people. A severe deficiency causes goiter and hypothyroidism because, despite an increase in thyroid activity to maximize iodine uptake and recycling, iodine concentrations are insufficient to enable synthesis of thyroid hormones [17].

### 2.2. Iron

Deficiencies are a public health problem worldwide [18]. In the year 2000, it was estimated that iron deficiency anemia affected two billion people, mostly women and children. Anemia concerns roughly a third of the world’s population [19], but anemia represents the end stage of an iron deficiency [20] which is largely prevalent in all categories of society [21]. The diagnosis of this deficiency is complex in acute and chronic diseases presenting an inflammatory response that modulates blood iron concentrations [20,22]. Its efficient treatment has often been prevented by beliefs such as the risk of iron administration causing additional oxidative stress or favoring infection. These concerns have not been confirmed [23]. The exploration of the iron status (deficiency < 7.1 µg/L or 12.5 mmol/L) included determining the presence of inflammation reflected by C-reactive protein (CRP) > 10 mg/L [24]. In addition to hemoglobin, and erythrocyte morphology (microcytosis, hypochromia), the following have been required: Serum ferritin (<30 ng/L), transferrin saturation (<15%), total iron-binding capacity (>13.1 µmol/L), soluble transferrin receptor (increased > 28.1 nmol/L) [20], and hepcidin (ranges men 0.6–23.3 nmol/L, women 0.4–19.7 nmol/L) [25].

With the availability of hepcidin as a marker of deficiency, the diagnosis should become easier [20,22,25]. Hepcidin is a hormone synthesized in the liver, secreted into the blood that systemically controls the rate of iron absorption as well as its mobilization from stores [26]. The synthesis of hepcidin is up-regulated by inflammatory cytokines (particularly interleukin-6), irrespective of the total level of iron in the body. This relationship most likely accounts for the development of anemia of chronic disease [27,28] (please see Section 3.7 for specificities in kidney diseases).

### 2.3. Selenium

Deficiencies have been an issue in Europe and in other parts of the world as the main source of human exposure is diet, which is related to the soil content [29]. Scientists have claimed that changes in climate and the organic carbon content of soil will lead to overall decreased soil Se concentrations, particularly in agricultural areas [30]. Europe and some parts of Australasia are particularly affected, while North America is spared by elevated soil content. Borderline Se status reduced endogenous antioxidant defenses via the reduction of the activity of the glutathione peroxidase (GPX) family of enzymes [31], which contributed to poor outcomes facing critical illness, and particularly sepsis [32]. In major trauma, the intravenous administration of selenium in combination with other micronutrients at 5 times DRI doses (see Table 1) resulted in normalization of plasma GPX3 activity and shortened hospital stays [33]. Nevertheless, the high dose single selenium trials (1000 µg/iv/day) in septic shock have been negative [34]. Therefore this strategy should not be considered, and has been advised against in the Surviving Sepsis Campaign 2017 recommendations [35].

### 2.4. Zinc

Deficiencies were first identified in the 1960s [36], and have continued to plague multiple regions of the planet [37]. It has affected all ages, but particularly the elderly in western regions. Zinc deficiency affects cell-mediated immune dysfunction, susceptibility to infections, and increases oxidative stress. A randomized trial including 50 healthy elderly subjects tested the impact of a zinc-supplement (45 mg elemental Zn/day) orally for 12 months versus placebo [38]. First, the authors showed that compared to younger, healthy subjects, the elderly study subjects had lower plasma zinc, higher ex vivo generation of inflammatory cytokines and interleukin 10. The supplement resulted in a significant reduction of the incidence of infections and ex vivo generation of tumor necrosis factor alpha and plasma oxidative stress markers compared to the placebo group [38].

### 2.5. Copper

Deficiencies have also been shown to affect specific geographic areas—drinking water may or may not be rich in copper depending on the pipe composition. The deficiency is most frequently acquired [39], e.g., due to insufficient intakes in vulnerable populations, increased demands (pregnancy, lactation, wound healing), malabsorption (including high phytic content of vegetarian diets), increased losses (e.g., major burns, continuous renal replacement therapy), and from hereditary diseases [40,41].

### 2.6. Vitamin A

Deficiency is considered the world’s most important cause of preventable blindness [42]. It affected 2.8 million children under five years of age. Tragically, the numbers have only grown over the last decades [20].

### 2.7. Vitamin D

Long known only for its role in calcium and bone homeostasis, and the development of osteoporosis, its deficiency seems to be a worldwide problem, and to particularly affect inpatients. The multiple effects of vitamin D are mediated by genomic and non-genomic effects, and include muscle function and metabolism, innate and adaptive immune system, lung epithelial function, cardiac function and numerous other functions as the specific nuclear vitamin D receptor are widely expressed throughout the body [43]. As shown in Table 1, low values can be found in nearly any severe condition. Many observational studies have consistently shown an association between low vitamin D levels and poor clinical outcomes. Nevertheless, high-quality evidence showing the benefits of vitamin D supplementation in inpatients is still lacking [44]. One of the largest studies (VITdAL-ICU) was analyzed retrospectively: 475 critically ill patients with 25(OH)D levels < 20 ng/mL [45]. The deficiency was not associated with persistent critical illness, nor did supplementation with vitamin D3 mitigate the development of persistent critical illness. The actual evidence does not support general vitamin D screening and supplementation for the medical inpatient population in an acute care setting [44]. By contrast, screening of chronic kidney disease patients is probably rational (see Section 3.7).

## 3. Disease Specific Deficiency

The below Table 2 summarizes the most frequent deficits encountered in the different pathologies.

### 3.1. Alcoholism

Micronutrient deficiencies are commonly encountered in alcoholic patients, not only explained by a decrease of global dietary intake, but also because of maldigestion, malabsorption, impaired hepatic activation and an increased breakdown and excretion. The risk of developing micro and macronutrient deficiencies has been known to increase significantly when alcohol makes up more than 30 percent of total caloric intake [46].

All fat-soluble vitamins (A, D, E and K) are susceptible to be decreased [47], although vitamins A and K deficiencies are more common in the case of overt hepatic disease or chronic pancreatitis. Among water soluble vitamins, the vitamin B1 (thiamine) deficiency has been the most frequently described and feared [48], potentially leading to Wernicke’s encephalopathy and its well-known triad (delirium, oculomotor abnormalities and ataxia). If left untreated, it can progress to the amnestic-confabulatory syndrome called Korsakoff. The therapeutic benefit of thiamin has been demonstrated in alcoholic patients even without severe Wernicke-Korsakoff encephalopathy [49]. In the emergency department, patients with acute alcohol intoxication have not all suffered thiamine deficiency [50,51]. A systematic review showed that thiamine and vitamin C were the most frequently identified deficiencies [51,52]. Nevertheless, considering the elevated prevalence of malnutrition in these patients, it is cautious and cheap, in case of hospital admission to provide 100–300 mg thiamine prior to any glucose IV infusion to prevent precipitating Wernicke’s encephalopathy. Excessive alcohol consumption has also been linked to zinc and copper deficiency, which seems to be associated with a decreased quality of physical and mental life [53]. 

There has been one randomized trial testing the administration of 600 mg of benfotiamine in alcohol dependent patients—this thiamine analog was associated with a reduction of alcohol consumption, but no metabolic variables were tested [54]. In the absence of other randomized trials, most of the experts advocated the empiric administration to alcohol dependent patients of multivitamin cocktails, including in particular, thiamine (200–300 mg), folic acid, vitamin B6, and vitamin C upon admission. 

### 3.2. Anemia

A third of the world’s population is affected by anemia, and iron deficiency is involved in 50% of the cases [18]. The most common symptoms are paleness, fatigue, dyspnea and headache. Laboratory tests have shown low blood haemoglobin concentration (Hb < 130 g/L in men, <120 g/L in women and <110g/L in pregnancy) with microcytosis, hypochromia and serum ferritin below 30 µg/L. Iron deficiencies can be physiological (e.g., pregnancy), or pathological in case of blood loss (e.g., surgery, trauma, digestive tract bleeding), malabsorption (e.g., celiac disease, gastrectomy), chronic disease (cancer, chronic heart failure) or in some genetic disorders [19]. In the presence of iron-deficiency anemia, investigations to identify a cause of blood loss or malabsorption are required. Preventive iron supplementation should be prescribed in at-risk patients using the oral route, while treatment of in patients with a positive diagnosis [19], may require the intravenous route due to the frequent gastric intolerance and poor absorption of oral supplements. 

Macrocytic anemia can be observed in vitamin B12 or/and vitamin B9 (folate) deficiencies. Vitamin B12 deficiencies occur in cases of severe malabsorption (bariatric surgery, gastrectomy, or autoimmune gastritis), in the abuse of nitrous oxide, and in cases of inherited metabolic disorders. Vitamin B12 supply is recommended after confirmed diagnosis. This treatment will be lifelong when the etiology of the deficiency is irreversible or unknown [55]. 

Copper deficiencies impair the activity of hephaestin, a copper-dependent ferroxidase responsible for transporting dietary iron from intestinal enterocytes into the circulatory system—its depression leads to iron deficiencies and low hemoglobin. Copper deficiency anemia has been treated with oral or intravenous copper supplementation [56]. Cobalt, a component of hydroxycobalamin, is considered one the most stimulator of erythrocyte production. In clinical practice, Cobalt administration is rare considering the high risk of toxicity of cobalt salts in humans [57].

### 3.3. Cardiomyopathies and Heart Failure

According to the European Society of Cardiology (ESC), cardiomyopathy (CM) is defined as a myocardial disorder in which the heart muscle is structurally and functionally abnormal, in the absence of coronary artery disease, hypertension, valvular disease and congenital heart disease [58]. CM can lead to heart failure (HF). Deficiencies in thiamine and selenium are nutritional factors that may be involved in the occurrence of such myocardial disorders [58,59] in malnutrition conditions, malabsorption, or exclusive parenteral nutrition (PN) [60,61,62]. Selenium deficiency cardiomyopathy is known as the Keshan disease in humans. It was initially described in China in 1935 in selenium-poor soils, with multiple case reports. In addition to the low plasma selenium levels, low glutathione peroxidase-1 activity has been reported in animals [63]. Some cases reported have been reversible by short-term oral or IV administration, and others have been fatal [60,61,62,63].

Thiamine plays a fundamental role in cellular metabolism, especially in the carbohydrate pathway. Severe and chronic thiamine deficiencies are known as Beriberi disease. The classic presentation includes neurologic features and encephalopathy. HF symptoms have been less frequent and have been associated to metabolic acidosis and hyperlactatemia (due to an inhibition of the pyruvate deshydrogenase) [64,65]. A fulminant form has also been described [64]. It has been shown that about 40% of the patients hospitalized for HF presented thiamine deficiencies [66]. Animal models have demonstrated that this deficiency causes cardiac disorders (cardiac hypertrophy, depressed cardiac contractility, and dysrhythmias) in the absence of beriberi. Finally, in human studies, the benefit of thiamine supplements in cases of chronic HF are unclear [66].

Iron deficiencies in HF are common (about 30–50%) in patients with chronic HF, independent of anemia, and have lead to skeletal muscle dysfunction. Recent ESC guidelines recommended screening for iron deficits in HF patients [28,59].

Recently, an association between vitamin D deficiency and HF has been suggested [67]. Vitamin D promoted cardioprotection in animals (anti-inflammatory, anti-apoptotic and anti-fibrotic mechanisms) [67].

### 3.4. Inflammatory Bowel Disease

Malnutrition is present in the vast majority of patients with inflammatory bowel disease (IBD), the deficiencies being more prevalent in Crohn’s disease compared with ulcerative colitis and more important in active diseases [68]. Micronutrient deficiencies are essentially explained by the reduced dietary intake and the underlying malabsorption. They have been associated with prolonged hospitalization and higher mortality [69]. 

Fat-soluble vitamins are particularly prone to deficiency. A high prevalence of vitamin A and E deficiencies have been reported [70], but also of vitamin D, which has been suspected to play a role in the pathogenesis of IBD [71]. Vitamin K deficiencies are also frequent and correlated with disease activity [72]. 

Folate deficiencies are common in IBD and aggravated by treatment, such as sulfasalazine or methotrexate. Vitamin B12 has also been frequently observed, especially in Crohn’s disease and after ileal resection of ≥30 cm [73]. Thiamin deficiency has also been reported, especially in IBD patients treated with parenteral nutrition [74].

Among trace elements, selenium deficiencies have been reported and may increase the severity of gut inflammation—the repletion data are conflicting [75]. Zinc deficiencies are also prevalent [76]. Iron deficiencies are very frequent and are the leading cause of anemia in patients with IBD. In the presence of inflammation, this diagnosis can be challenging and relies on the values of serum ferritin values. 

### 3.5. Liver Disease

The liver plays a crucial role in maintaining systemic Zn homeostasis [77]. Chronic liver disease, such as chronic hepatitis, liver cirrhosis, or fatty liver, impairs Zn metabolism, and has resulted in Zn deficiency, which in turn has caused multiple metabolic abnormalities, including insulin resistance, hepatic steatosis and hepatic encephalopathy. Zn deficiency may also favor carcinogenesis of hepatocellular carcinoma (HCC). In chronic liver disease, low levels of selenium [78] have been generally observed. In comparison, copper levels have often been elevated [79]. It is argued that doses required to achieve an effect in chronic hepatitis are far beyond DRI with 150–200 mg/day [80].

In cases of liver disease secondary to alcoholism, please see 3.1 (same picture). Group B vitamin deficiencies, especially thiamine, are common in cirrhosis [81]. Unlike observations for alcoholic patients without liver disease, vitamin B12 levels have been frequently elevated in viral hepatitis, cirrhosis and hepatocellular carcinoma (HCC) [82]. This increase has been explained by the cytolysis of hepatocytes, and vitamin B12 being mainly stored in the liver. 

Fat-soluble vitamin deficiencies have been observed in cases of alcoholism, but also in cholestasis with malabsorption and bile salt deficiency [83]. Levels of vitamins A, D and E should therefore be routinely checked as well as prothrombin time. However, prolonged prothrombin time does not purely reflect vitamin K deficiencies, but also reduced levels of coagulation factor V. Importantly, vitamin D has pleiotropic effects for liver disease, including anti-inflammatory, immune-modulatory and anti-fibrotic properties in addition to its classical skeletal effects. This may impact on disease progression [84], especially in HCC and non-alcoholic steatohepatitis, although any benefit from its repletion has not been formally proven by prospective studies [85].

### 3.6. Obesity & Bariatric Surgery

In patients with grade III obesity (body mass index ≥ 40 kg·m^−2^ or ≥ 35 kg·m^−2^ with comorbid conditions), bariatric surgery has become common. In 2013, over 450,000 bariatric surgeries were performed worldwide. Roux-en-Y gastric bypass, adjustable gastric band, and sleeve gastrectomy were the most frequent procedures [86,87]. Micronutrient deficiencies were observed before (in obese patients, vitamin C deficiency in about 40% and zinc deficiency is up to 50%) and after the surgery. Due to fat malabsorption and maldigestion, all fat-soluble vitamins are at risk. The bypass of the duodenum and proximal jejunum lead to thiamine deficiencies [86], leading to a risk of clinical Wernicke’s encephalopathy [88]. Vitamin B9 is approximately 10%. Vitamin B12 deficiencies have been widely described in the literature because of their neurologic complications, especially an acquired myelopathy with paresthesias, ataxia and muscle weakness [89]. It is recalled that the intrinsic factor produced in the stomach is needed for ileal absorption of vitamin B12. The severity of a vitamin B6 deficiencies vary from peripheral neuropathy to seizures [89]. The common trace elements deficiencies after bariatric surgery include copper (absorbed in the stomach and the duodenum), iron (absorbed in the duodenum), and zinc (absorbed in the jejunum). The prevalence of copper deficiency has been reported to be as high as 90% post- surgery (70% for zinc). Further, systematic supplements of micronutrients are recommended after bariatric surgery by the American Society for Parenteral and Enteral Nutrition (ASPEN) [86] and the American Society for Metabolic and Bariatric Surgery clinical practice guidelines (ASMBS) [90]. 

### 3.7. Kidney Disease

Kidney disease, whether chronic (CKD) or acute, affects micronutrient homeostasis and might lead to either deficiency or toxic excess. Indeed, a decreased glomerular filtration rate might lead to accumulation of molecules normally excreted by the kidney such as selenium. On the other hand, renal replacement therapy when applied might lead to uncompensated losses of other micronutrients (copper). Finally, the loss of renal activation might lead to decreased biological activity (vitamin D). Unfortunately, the authors knowledge remains limited [91].

Anemia is commonly observed in CKD. Beyond the lack of erythropoietin, which is now clearly established and easily administered, it might also be associated with alterations in iron metabolism and inflammation. Oral corrections of iron deficiencies in pre-dialysis CKD patients have been shown to be an efficient option in a recent randomized trial [92]. The role of hepcidin, a key regulator of circulating iron level in CKD associated anemia is increasingly recognized. Indeed, the condition has been associated with elevated hepcidin serum levels as it is typically excreted by the kidney [93]. This leads to reduced iron availability and anemia. Its measurement and potential anti-hepcidin therapies could help managing anemia in CKD [27,94]. 

Subclinical vitamin K deficiencies have been shown to be clinically relevant as requirements have increased due to the vitamin K-dependent proteins required to inhibit calcification [95]. This vitamin governs the gamma-carboxylation of matrix Gla protein for inhibiting vascular calcification, and the vitamin D binding protein receptor is related to vitamin K gene expression [96]. Deficiency may favor vascular calcification.

Chronic dialysis is typically associated with elevated oxidative stress leading to low levels of zinc, selenium and GPX. A French team showed three decades ago that weekly administration of selenium with zinc was able to restore GPX activity and reduced thiobarbituric acid reactants (TBARs) plasma concentrations [97]. In a large cohort of 1278 patients on incident hemodialysis, it was observed that lower selenium and zinc concentrations were strongly and independently associated with death and all-cause hospitalization [98]. Trimestrial monitoring of selenium and zinc may thus be justified, with repletion in case of low values. 

In acute kidney failure requiring continuous renal replacement therapy, other micronutrients such as thiamine and copper will be lost in the effluent fluid [39,40,99]. Copper losses causing low blood levels can be associated with severe arrhythmias and wound healing complications. Very low plasma levels (<8 mmol/L) might require active intravenous repletion with doses 6–10 mg/day, i.e., 5–10 times the usual DRI [40].

### 3.8. Migrant Populations

Being a migrant is not a disease, but a difficult social condition frequently associated with malnutrition for multiple reasons ranging from insufficient food intake, to exposure to unusual food or unbalanced diet due to incapacity to find the traditional foods. Migrants represent a growing category of inpatients. Micronutrients of concern have shown to be retinol, vitamin D, magnesium, potassium, copper, and selenium [11], in addition to iron.

### 3.9. Laboratory Investigations

Laboratory investigations of micronutrient deficiencies have often not been systematic except for 2 conditions: (1) Anemia outwork, which generally included determination of blood Vitamin B6, B12, iron, ferritin and transferrin; (2) screening before and the follow-up after bariatric surgery included thiamine, vitamin B12, folic acid, iron, zinc, copper, calcium and the liposoluble vitamins D, A, E and K were recommended [90]. In other conditions, the diagnosis lacked standardization. A pragmatic approach upon admission or during hospitalization could be to draw an additional blood sample for further diagnostic outwork, and to empirically administer multi-micronutrients without delay.

Inflammation causes a redistribution of micronutrients between the various compartments and generally reduces circulating levels. The intensity of inflammation has been reflected by CRP levels [24]. The example of vitamin D has been emblematic. While a CRP > 80 mg/L has been associated with a reduction of its blood concentration by 40% below reference ranges [24], CRP has never been mentioned in the vitamin D trials. CRP should belong to any micronutrients outwork as low levels do not necessarily indicate a deficiency.

## 4. Micronutrient Unavailability as Cause of Deficiency

### 4.1. Nutritional Sources

Hospital related malnutrition is a well know entity that is observed worldwide and is related to being bedridden [100]. A large proportion of patients only consume a third or half of the proposed meals [4]. In our hospital, a standard daily serving provides 1700–1800 kcal. If only half is consumed, the daily micronutrients cannot be covered.

The sickest patients are fed with enteral nutrition. Due to regulatory constraints, the industry must respect micronutrient recommendations intended for the general healthy population, the previously mentioned DRI. The concentrations of the products are calculated for feeding doses varying between 1500 and 2500 kcal/day. However, worldwide it has been shown that many patients receive no more than 1000 kcal/day by this route [101]. In addition, the most recent nutrition guidelines for critical care patients recommended ramping up the feeding over several days [102]. The consequences might be an even further reduction of nutrition delivery. By design, the enteral feeding solutions will not be able to cover needs as long as quantities below 1500 kcal per day are provided. Table 3 shows the detailed micronutrient provision for 1000 kcal/day provided by the 10 most frequently used enteral feeding solutions provided by 4 international companies on the Swiss market. While intakes below DRI of fluor and iodine may be less important during an acute phase, low intakes of iron and of the B vitamin group are a concern considering their essential role in energy (ATP synthesis) and carbohydrate metabolism. Moreover, as absorption is unreliable and as needs might be higher, several micronutrients such as vitamin C are just in the reference range, which may be insufficient.

Regarding these low micronutrient doses, it might be justified to deliver standard multivitamin and trace element products daily providing DRI doses to the majority of inpatients. This strategy has been applied for many years in the Lausanne university hospital’s ICU, as critically ill patients do have higher needs. In the sickest patients, multi-micronutrients cocktails have been delivered IV for the first 5 days [33], resulting in a shortening of the hospital stay, particularly in major trauma patients.

### 4.2. Geriatric Population

The qualification, elderly, encompasses patients aged 60 to over 100 years. While the younger seniors often are fit until the seventies, some physiological changes already occur that become exacerbated with growing age. The elderly often present with anorexia which is considered a complex geriatric syndrome and a risk factor for frailty [9]. These changes are associated with lower weight, and lower energy expenditure. The body undergoes specific changes—the gastric mucosa tends to atrophy, reducing vitamin B12 absorption. Indeed, the decline in vitamin B12 is independent of nutrition but caused by a decline of both the intestinal uptake and the renal reabsorption system for vitamin B12 [103]. The elderly also require higher doses of vitamin B6 and D to maintain health, which has been integrated into DRI recommendations for older subjects. Deficiencies can be overcome by supplementation, as shown by a large randomized controlled trial including 652 geriatric patients—daily oral nutrition supplements enriched with proteins, hydroxy-methyl-butyrate, vitamin D and other micronutrients reduced mortality [104].

### 4.3. Partial or Complete Starving upon Hospital Admission and Refeeding

The incidence and importance of the refeeding has often been underestimated [105]. The absence of a uniform definition participates in its underestimation [105]. Refeeding syndrome consists of metabolic changes that occur on the reintroduction of food or simply a glucose infusion. A few days of feeding grossly below needs will be sufficient to create the metabolic crisis which is characterized by sudden shifts in the electrolytes that are needed for energy and substrate (mainly glucose) metabolism. The NutritionDay survey, an initiative that analysed the relation between nutritional intakes and outcomes of a wide range of institutionalized and hospitalized patients worldwide, has shown that more than half of the patients admitted to hospital were eating less than half of their normal food intake before admission [106]. This places the majority of hospitalized patients at risk of a refeeding syndrome. 

Some categories of patients, such as chronic, alcohol consumers, which are largely prevalent in western countries, being present in nearly 30% of hospital admissions, are at higher risk of thiamine deficiencies—these patients are at particularly high risk of refeeding syndrome and its worst neurologic complication, the Wernicke encephalopathy. A recent review of the literature confirmed the importance of administering intravenous thiamine to these patients in order to prevent severe sequelae [107]. The recommended doses ranged from 50–100 mg/day to 250–500 mg 3 times a day. The IV route was recommended due to the frequent presence of gastritis in these patients, which reduced absorption.

### 4.4. Economic Considerations

A Canadian prospective cohort study showed that approximately 40% of the 956 patients admitted to hospital were moderately to severely malnourished. These patients had longer hospital stays, and as a result, cost more than the well-nourished patients [108]. A European narrative review showed that malnutrition increased the length of hospital stays by 2.4 to 7.2 days [5]. Malnutrition led to an additional individual cost ranging between 1640 € and 5829 €. 

Clinical evidences are lacking for empirical multi-micronutrient supplements. The cost of malnutrition attributable to micronutrient deficiencies have not been assessed in adult inpatients and several studies have failed to demonstrate significant beneficial effects of various micronutrient supplements in the general population [109]. Nevertheless, there are data suggesting a benefit, at least in the sickest patients [33,110]. 

However, data exists for children. A study focusing on estimates of disability-adjusted life years and their monetization showed that short-term economic costs of micronutrient malnutrition in India amounted to 0.8% to 2.5% of the gross domestic product [14]. The health and cost consequences of iodine, iron, vitamin A, and zinc deficiencies were assessed in Pakistani children: Societal costs amounted to 1.44% of gross domestic product and 4.45% of disability-adjusted life-years in Pakistan in 2013, which hindered the country’s development [111].

When deciding about an empirical administration of micronutrients, the analytical costs of deficiency diagnosis must be considered. The European Society for Clinical Nutrition and Metabolism (ESPEN) monitoring recommendations indicate that some vitamin and trace element analysis (inductively coupled plasma mass spectrometry: ICP-MS) are expensive [112]. They are actually more expensive than the empirical administration. A semi-automatic weekly determination of blood selenium levels in our ICU resulted in major costs that could be contained by the decision to let blood sampling be prescribed only by the ICU nutritionists and dieticians in patients at risk [113]. 

A multi-micronutrient tablet costs 0.80 € (the IV dose in Table 1 costs 25 €), which is negligible compared to the cost of one day in hospital, or worse, in the ICU. This prescription is likely to be beneficial if the administration is standardized, and limited to the first week of hospitalization, and to patients on enteral feeding.

## 5. Conclusions

Micronutrient deficiencies and borderline status are more frequent than generally acknowledged. The most important potential acute deficiency that may compromise outcome is thiamine deficiency. Other deficiencies will impact on immune defenses and anabolic capacity. Therefore, an empirical and cheap complementation strategy, based on daily oral multi-micronutrient products providing DRI, may be justified for hospital inpatients for one week. It is important to state that the evidence from trials is still missing.

## Figures and Tables

**Table 1 jcm-08-00931-t001:** Micronutrient strategy in critically ill patients admitted to the Lausanne multidisciplinary ICU, according to disease and nutrition therapy.

Situation	Stress Profile in High Risk Patients in Organ Failure *	Parenteral Nutrition (and Combined Feeding)	Enteral Nutrition
Micro-Nutrients	1 vial multi-trace element (Addaven^®^, Fresenius Kabi, Oberdorf, Switzerland) + 5 mg Zinc + 1 vial multi-vitamin (Cernevit^®^, Baxter, Volketswil, Switzerland) + 500 mg vitamin C + 100 mg vitamin B1	Same as stress profile	Multi-micronutrient providing DRI needs (Supradyn^®^, Roche, Basel, Switzerland)
Duration Route	Diluted in 100 ml de NaCl 0.9% over 6 hours from admission for first 6 days during night shift	Daily with parenteral nutrition	Daily Mixed with enteral feeding

*: High risk conditions include shock (cardiogenic, septic, hypovolemic), pancreatitis, severe hepatopathy, major trauma, organ transplant, and malnutrition.

**Table 2 jcm-08-00931-t002:** Disease specific vitamins and in trace elements deficiencies.

Disease	Micronutrients at Risk
Alcoholism	Zn Vitamins A, D, E, K, B12, B9, B6, B1, B2, C
Anemia	Fe, Cu, Co Vitamins B12, B9
Cardiomyopathies/ Heart failure	Se, Fe Vitamin B1, D ^?^
Inflammatory bowel diseases	Se, Zn Vitamins B12, A, D, E, K
Liver diseases	Se, Zn Vitamins B12, A, D, E
Obesity and Bariatric surgery	Cu, Zn, Fe Vitamins A, D, E, K, B1, B9, B12, C
Kidney diseases (chronic & acute)	Chronic: Vitamins K, D Acute: B1, Fe, Se, Zn, Cu

^?^: means uncertainty as to deficiency.

**Table 3 jcm-08-00931-t003:** Energy, protein and micronutrient data for 1000 kcal for a selection of frequent products available on the Swiss market compared to the dietary reference intakes (DRI). The values which are below the DRI appear in red and bold characters; in violet-bold, those for which DRI is just covered.

	Abbott	Abbott	Nestlé	Nestlé	Nestlé	Fresenius K	Fresenius K	Fresenius K	Nutricia	Nutricia	
Values for 1000 kcal	Promote Fibres Plus	Jevity Plus	NovaSource GI Advance	Isosource Energy	Peptamen Intense	Fresubin 2 kcal HP	Fresubin HP Energy	Fresubin Intensive	Nutrison Protein + Mulitf	Nutrison	DRI Adults
**Energy** density kcal/ml	1.3	1.2	1.55	1.57	1.0	2.0	1.5	1.2	1.3	1.0	
**Proteins** g/1000 kcal	62.5	46.3	61.9	38.9	93.0	50.0	50.0	83.3	49.2	40.0	
Fer (Fe) mg	**12.3**	**15.0**	**11.0**	**10.2**	**16.0**	**13.5**	**8.7**	**16.7**	**15.6**	**16.0**	18
Zinc (Zn) mg	13.1	11.7	11.6	9.6	13	12	**8.0**	12.5	11.7	12	8
Cuivre (Cu) mg	1.5	1.7	1.5	1.5	1.8	1.5	**0.7**	1.7	1.8	1.8	0.9
Manganèse (Mn) mg	3.1	3.5	2.3	2.3	1.4	2.5	2.0	4.2	3.2	3.3	1.8
Fluor	**0.0**	**0.0**	**1.0**	**1.3**	**1.6**	**1.5**	**0.7**	**1.7**	**1.0**	**1.0**	3
Iode (I) μg	**123**	**125**	**142**	**146**	**120**	**134**	**89**	183	**102**	**130**	150
Molybdène (Mo) μg	92	108	116	115	170	100	67	117	102	100	45
Chrome (Cr) μg	54	67	97	96	60	67	45	92	65	67	20
Selenium (Se) μg	65	63	65	64	80	67	**45**	88	**55**	**57**	55
A (RE) μg	1154	**700**	1097	1083	**650**	925	**613**	1500	**797**	820	750
D μg	**6.9**	**8.3**	14.2	14	14	**10**	**8.7**	17	**13.3**	**10**	10
E (α-TE) mg	18.2	20	17.4	16.6	**14**	**13.5**	**8.7**	25	**12.5**	**13**	15
K μg	**54**	**67**	**71**	**76**	**44**	**67**	**45**	**75**	**52**	**53**	90
C mg	154	100	123	102	80	**67**	**45**	183	102	100	75
B1 mg	1.5	1.6	1.6	1.5	**1.0**	1.5	**0.7**	1.7	1.5	1.5	1.1
B2 Riboflavin mg	2.2	1.8	1.7	1.7	1.3	2.0	1.3	1.7	1.6	1.6	1.1
B3 Niacin mg	21.5	18.3	20.0	17.2	30	**16**	**11**	20	18	18	14
B5 Pantothenic acid mg	7.7	8.3	**5.2**	5.5	**4.5**	**4.5**	**3.3**	7.5	**5.2**	**5.3**	5
B6 Pyridoxin mg	2.2	2.2	1.8	1.8	1.7	**1.5**	**0.8**	2.5	**1.6**	**1.7**	1.5
B12 Cyancobalamin μg	4.6	2.9	3.8	3.7	2.9	**2.5**	**2.0**	4.2	**2.0**	**2.1**	2.4
B9 Folic acid μg	**231**	**250**	**290**	**287**	**300**	**267**	**180**	**263**	**258**	**270**	400
B8 Biotin μg	46	43	45	45	**30**	50	**33**	57	**39**	40	30
Choline mg	**462**	500	**368**	**382**	670	**0**	178	**0**	**359**	**370**	425

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
