# Peer review of "Micronutrient Deficiencies in Medical and Surgical Inpatients"

_jcm, 2019, doi:10.3390/jcm8070931_

Round 1
Reviewer 1 Report
Authors made efforts to address the importance of micronutrients, and this is really a relatively under-attended toipic. This manuscript may help remind medical personnel of this subject.
Authors tried to cover a wide range of micronutrients-related clinical topics in this manuscript. Ths is in one way worth appreciation, but in another way, weakens the sceintific value. Many topics were mentioned, but were unable to be address in any details.
To strenthen this manuscript, authors may consider to make your suggestion or proposition about how to correct or prevent the micronutrients deficiency for inpatients.
Author Response
Reviewer 1 :
Authors tried to cover a wide range of micronutrients-related clinical topics in this manuscript. Ths is in one way worth appreciation, but in another way, weakens the sceintific value. Many topics were mentioned, but were unable to be address in any details.
The editors invited us to write a review about “micronutrients” in general which proved to be a difficult task. We understand the reviewer’s comment but it Impossible to go into details without writing a textbook. There are no “global inpatient” micronutrient data/studies available: the only way to provide objective data was to target pathologies where some data are available. The limited number of pathologies we have addressed are all relevant to the hospital world and generate a large part of admissions, and have micronutrient specificities and deficiencies. This has now been further completed by the inclusion of kidney disease (a request or R#2)
We tried to address an important issue which is the global underfeeding in hospital and the low micronutrient content of the enteral feeding solutions: this is the rational for providing multi-MN
To strenthen this manuscript, authors may consider to make your suggestion or proposition about how to correct or prevent the micronutrients deficiency for inpatients.
We provided a general recommendations in the conclusion of the first version: we showed that majority of patients on enteral nutrition do not receive sufficient MN doses to cover DRI- further majority of patients do not eat much as inpatients – these are our justifications for providing a daily multi-micronutrient to all “inpatients”. Aim is not to achieve fantastic goals, but to prevent deterioration.
Reviewer 2 Report
This review article contains important information about inpatient micronutrient deficiency and is helpful to many clinicians.
I think add of additional information can make this review article more helpful to international readers.
Micronutrients deficiencies in chronic kidney disease are important problems and should be evaluated in this review article.
Line 89
More specific explanations will be helpful; methods, cutoff value, diagnostic criteria or suggestion about using hepcidin for diagnosis of anemia
Line 96
Several trials evaluated selenium supplement for critically ill patients. Results of these studies and recent consensus (i.e. surviving sepsis campaign) should be mentioned.
Line 152
There are some debates and should be mentioned. Studies for acutely ethanol-intoxicated ED patients had alcohol-dependent homeless populations showed a low prevalence of thiamin deficiency. (Li SF. 2008, Lee HJ. 2015, Ijaz S. 2017).
Thiamin supplement before glucose infusion has reasonable pathophysiologic explanations. However, clinical evidence is lacking. Moreover, it can delay the treatment of symptomatic hypoglycemia and confusing – only 1st day of admission? After correction of thiamin deficiency? After 100mg loading?
Line 305
don not -> do not?
Line 333
I think the detailed introduction of your protocol can be helpful for many clinicians.
Line 374
I agree with your opinion. However clinical evidences are lacking for empirical vitamins supplements. Many studies have failed to demonstrate the beneficial effects of various vitamin supplements. Evaluation of the cost-benefit effect of vitamin supplement should be covered in this article.
Author Response
Reviewer 2
This review article contains important information about inpatient micronutrient deficiency and is helpful to many clinicians.
Thank you for your positive comments and suggestions J
I think add of additional information can make this review article more helpful to international readers.
Micronutrients deficiencies in chronic kidney disease are important problems and should be evaluated in this review article.
Thank you for suggestion. We have added a paragraph devoted to kidney diseases (please see § 3.7)
Line 89: More specific explanations will be helpful; methods, cutoff value, diagnostic criteria or suggestion about using hepcidin for diagnosis of anemia
Iron deficiency diagnostic criteria have been included in the paragraph
Line 96: Several trials evaluated selenium supplement for critically ill patients. Results of these studies and recent consensus (i.e. surviving sepsis campaign) should be mentioned.
Indeed, high doses isolated selenium was not proven efficient, and should not be provided (reference has been inserted). Nevertheless, as written selenium poor status is an issue in Europe, and this element should be provided but in doses not exceed 5 times the DRI along with other micronutrients – not as solo element to respect the endogenous antioxidant system. Our own data (RCT) show benefit of this strategy (Berger et al 2008): this is why we ended using this “cocktail” which we have included as Table 3. Its composition is based on our trial.
Line 152: There are some debates and should be mentioned. Studies for acutely ethanol-intoxicated ED patients had alcohol-dependent homeless populations showed a low prevalence of thiamin deficiency. (Li SF. 2008, Lee HJ. 2015, Ijaz S. 2017).
Thiamin supplement before glucose infusion has reasonable pathophysiologic explanations. However, clinical evidence is lacking. Moreover, it can delay the treatment of symptomatic hypoglycemia and confusing – only 1st day of admission? After correction of thiamin deficiency? After 100mg loading?
The 3 references have been inserted. The paragraph has been modified accordingly mentioning the debate. Indeed the evidence is limited and not all patients do present with deficiency, but considering the human and medical costs associated with encephalopathy and the costs of laboratory determinations, versus the very low costs of thiamine administration, we maintain the sentence: the dose of 100 mg is the minimum, most centers using 300 mg which has been mentioned.
Line 305 : don not -> do not?
Corrected - thank you
Line 333 :I think the detailed introduction of your protocol can be helpful for many clinicians.
Thank you for suggestion. We have inserted our protocol as Table 3.
Line 374 I agree with your opinion. However clinical evidences are lacking for empirical vitamins supplements. Many studies have failed to demonstrate the beneficial effects of various vitamin supplements. Evaluation of the cost-benefit effect of vitamin supplement should be covered in this article.
Thank you for supporting our conclusion, and the suggestion to include cost benefit consideration. It was not easy to realize: while hospital malnutrition costs have only been partially assessed, the MN part of malnutrition has not been addressed specifically in adult inpatients: there are solid data in children though. We have attempted to provide a balanced discussion of this issue (please see 4.4.)
J - thank you for the suggestion!
Round 2
Reviewer 2 Report
Concerns are well covered.
Thank you for making a good reference on micronutrient deficiencies.